# Effects of Germination and Popping on the Anti-Nutritional Compounds and the Digestibility of *Amaranthus hypochondriacus* Seeds

**DOI:** 10.3390/foods11142075

**Published:** 2022-07-13

**Authors:** Carmen Valadez-Vega, Olivia Lugo-Magaña, Claudia Figueroa-Hernández, Mirandeli Bautista, Gabriel Betanzos-Cabrera, Aurea Bernardino-Nicanor, Rosa María González-Amaro, Rodrigo Alonso-Villegas, José A. Morales-González, Leopoldo González-Cruz

**Affiliations:** 1Área Académica de Medicina, Instituto de Ciencias de la Salud, Universidad Autónoma del Estado de Hidalgo, Ex-Hacienda de la Concepción, Tilcuautla, San Agustín Tlaxiaca C.P. 42080, HG, Mexico; 2Preparatoria Número 1, Universidad Autónoma del Estado de Hidalgo, Av. Benito Juárez S/N, Constitución, Pachuca de Soto C.P. 42060, HG, Mexico; 3CONACYT-Tecnológico Nacional de México, Campus Veracruz, Unidad de Investigación y Desarrollo en Alimentos, M. A. de Quevedo 2779, Veracruz C.P. 94897, VE, Mexico; claudia.fh@veracruz.tecnm.mx; 4Área Académica de Farmacia, Instituto de Ciencias de la Salud, Universidad Autónoma del Estado de Hidalgo, Ex-Hacienda de la Concepción, Tilcuautla, San Agustín Tlaxiaca C.P. 42080, HG, Mexico; mirandeli@hotmail.com; 5Área Académica de Nutrición, Instituto de Ciencias de la Salud, Universidad Autónoma del Estado de Hidalgo, Ex-Hacienda de la Concepción, Tilcuautla, San Agustín Tlaxiaca C.P. 42080, HG, Mexico; gbetanzo@uaeh.edu.mx; 6Tecnológico Nacional de México/IT Celaya, Antonio García Cubas Pte #600 esq. Av. Tecnológico, Celaya C.P. 38010, Mexico; aurea.bernardino@itcelaya.edu.mx (A.B.-N.); leopoldo.gonzalez@itcelaya.edu.mx (L.G.-C.); 7CONACYT-Instituto de Ecología, A.C. INECOL 1975–2021, Carretera Antigua a Coatepec 351, Col. El Haya, Xalapa C.P. 91073, VE, Mexico; rosa.gonzalez@inecol.mx; 8Facultad de Ciencias Agrotecnológicas, Universidad Autónoma de Chihuahua, Av. Pascual Orozco s/n Campus 1, Santo Niño, Chihuahua C.P. 31350, CH, Mexico; ralonso@uach.mx; 9Laboratorio de Medicina de Conservación, Escuela Superior de Medicina, Instituto Politécnico Nacional, Plan de San Luis y Díaz Mirón, Col. Casco de Santo Tomás, Del. Miguel Hidalgo, Ciudad de Mexico C.P. 11340, Mexico; jmorales101@yahoo.com.mx

**Keywords:** *Amaranthus hypochondriacus*, anti-nutritional, in vitro digestibility, lectins, tannins, germination, popping

## Abstract

Amaranth seeds, although a valuable food in Mexico, contain anti-nutritional compounds that can affect food quality. As a part of this work, the proximate composition, fatty acid profile, protein digestibility, and the effect of germination and popping of *Amaranthus hypochondriacus* seeds was analyzed with the aim of eliminating anti-nutritional compounds. Untreated seeds comprised of 11.35–18.8% protein and 0.27–13.39% lipids, including omega 3, 6, and 9 fatty acids such as oleic, linoleic, linolenic, and arachidonic acid. The main minerals detected were Ca^+2^, K^+1^, and Mg^+2^. Nevertheless in vitro studies indicate that germination significantly improved digestibility, whereby treatments aimed at reducing anti-nutritional compounds decreased lectin concentration, while significantly increasing tannins and completely eliminating trypsins and saponins.

## 1. Introduction

Amaranth grains (*Amaranthus* spp.) are pseudo-cereal native to America, and were cultivated on thousands of hectares by pre-Columbian civilizations for use as food as well as in religious rituals. Some indigenous populations are known to have used amaranth grain, along with maize and beans, as an integral part of their cultivation plans. The Aztecs relied on amaranth seeds as an essential staple food. Amaranth (family Amaranthaceae), collectively known as amaranth, is a cosmopolitan genus of annual or short-lived perennial plants consisting of approximately 60 species. Uses for human consumption can be divided into the grain and vegetable amaranth [1].

It is well known that amaranth has excellent nutritional characteristics due to the high content of micronutrients such as calcium, magnesium, iron, vitamin C, and other nutrients such as carbohydrates necessary for a healthy life [2]. The nutritional value of amaranth has been extensively studied. It has been shown that amaranth leaves are excellent protein sources, with their maximum accumulation in the flowering phase [3]. Although simple to use, the traditional method of amaranth seed popping can result in a low recovery rate, as overheating leads to heterogeneous heating, browning, and carbonation. The nutritional value, such as the essential amino acids and vitamin content, as well as product quality, declines as a result [4]. To overcome these disadvantages, Murakami et al., 2014 developed a fluidized bed continuous processing system based on hot air heating for producing popped amaranth seeds in bulk [5]. 

Available evidence further indicates that amaranth seeds may contain some anti- and non-nutritional compounds. Antinutrients, commonly found in plant food, have both adverse effects and health benefit, is a substance occurring in the diet which acts antagonistically toward one or multiple nutrients, reducing bioavailability. This is usually done through complex formation which reduces nutrient absorption. Non-nutritive substances, which are not only unnecessary but may be harmful to the organism [6,7]. 

Extant studies focusing on toxic compounds in amaranth, including phenolic compounds—saponins, tannins, phytic acid, oxalates, protease inhibitors, nitrates, polyphenols, and phytohemagglutinins—indicate that oxalates and nitrates are the most significant concern when amaranth grain is destined for fodder. However, other compounds such as oxalic acid, phytates, saponins, and alkaloids that restrict the absorption of nutrients in food and their high accumulation can cause health problems. The findings reported by Valadez-Vega, further indicate that *Amaranthus hypochondriacus* species have high concentrations of lectins and trypsin inhibitors and lower concentrations of tannins and saponins [8]. The findings reported by Valadez-Vega, further indicate that *Amaranthus hypochondriacus* species have high concentrations of lectins and trypsin inhibitors and lower concentrations of tannins and saponins [9].

According to the results reported by Alegbejo, thermal processing of amaranth seeds aimed for human consumption may be a promising way to reduce the adverse effects of the aforementioned anti-nutritional and toxic factors [3]. Guided by this evidence, as a part of the present study, the effect of some traditional processes carried out in Mexico (such as popping and germination) on the persistence of anti-nutritional compounds were evaluated. The obtained results could be used to determine the most optimal way of processing amaranth seeds in order to improve their digestibility and nutritional value.

## 2. Materials and Methods

### 2.1. Plant Materials

Amaranth seeds (*Amaranthus hypochondriacus*) were obtained directly from the farmers in three Mexican States: Hidalgo, from Progreso (PR) and Mixquiahuala (MX1 and MX2) locations; State of Mexico (EM); and Puebla (PU). The seeds were cleaned, selected, and stored at 5 °C until use.

### 2.2. Chemicals

Standard multi-ionic solution for plasma spectrometry is from Perkin-Elmer, Waltham, MA, USA. N-benzoyl-DL-arginine, +catechin, p-nitroanilide, β-glucosidase, bacterial protease, porcine pancreatic trypsin, bovine pancreatic α-chymotrypsin, porcine intestinal peptidase, trypsin, boron trifluoride, and vainillin were from Sigma (Sigma Chemical Co., St Louis, MO, USA). All the other chemicals were of analytical grade.

### 2.3. Popping and Germination

Grains were separately processed by germinating and popping. The traditional popping procedure with some modifications was adopted, whereby 100 g of dry amaranth grain was placed on the claypan and was heated between 200–240 °C, on a whirlpool WF6559S gas cooker (Whirlpool Mexico) set to medium heat for about 1–2 min while stirring using a wooden ladle. Heating continued until the grain turned whitish, which typically required about 90 s. 

For germination, 20 g of seeds were washed with boiled water and left to soak for 12 hours, after which the water was removed, and a 5% sulfuric acid solution was added to soften the husk. After two hours, seeds were washed with distilled water before being placed in an oven at 35 °C and 100% relative humidity. The seeds were germinated by 72 h, rinsed daily with boiled water, and were considered germinated once the radical were 2.5–3 mm long, then were dried in an oven (Sheldon Manufacturing, Inc. Mod. Ex6 ZZMFG) at 25 °C.

A grinder Cole Parmer (model 4301-00) was used to grind unprocessed and processed grains into flour, them sieved (USA Standard testing Sieve).

### 2.4. Seeds Size and Weight

Amaranth dimensions (length, width, and thickness) were measured on fifty whole, healthy seeds each. Measurements were made with a Vernier caliper (Model No. 530-104, Mitutoyo, Kawasaki, Japan). The weight of 100 randomly chosen seeds was determined and reported as g/100 seeds.

### 2.5. Proximate Analysis 

Unprocessed amaranth seed flour was analyzed according to the AOAC method, whereby protein (920.87), using a factor of 6.25 to calculate the percentage of total protein), moisture (925.10), fat (920.39C), ash (923.03), and fibber (962.09) content were calculated, and the remaining percentage was ascribed to carbohydrates [10]. 

### 2.6. GC-MS Determination of the Fatty Acid Profile

Oil extraction was made following the AOCS Method (Ba 3–38.45) using petroleum ether and the extract was dried and stored at −80 °C until analysis. 

Fatty acid methyl esters (FAME) were prepared using a method based on a 14 % boron trifluoride–methanol solution, according to the AOAC method. The samples obtained were analyzed by Perkin Elmer Auto System XL spectrometer coupled to Agilent 6890 Gas Chromatograph using a fused silica capillary column (DB-Wax 30 m, 0.53 mm, 0.25 µm). The FAME peaks were identified by comparing their retention times of highly purified FAME standards [10].

### 2.7. Mineral’s Determination

Ground unprocessed (non-germinated) seeds were digested with 10 mL of concentrated nitric acid (J. T. Baker, CDMX, Mexico) in a Teflon TFM vessel with a Milestone Start D microwave digestion system (Model, Multiwave Anton Paar, Perkin-Elmer, Austria) at 75 bar pressure, 500 W power and 280 °C for 25 min. Next, the samples were filtered with Whatman paper of 0.4 μm pore size and were stored at 4 °C (Model, Multiwave Anton Paar, Perkin-Elmer, Waltham, MA, USA) with 75 bars of pressure and 500 watts of power at 280 °C for 25 min. Next, the sample were filtered with Whatman paper of 0.4-micron pore size and were stored at 4 °C [11]. 

The concentrations of chemical elements most commonly present in amaranth seeds (B^3+,^ Ca^2+^, Fe^2+^, K^+^, Mg^2+^, Mn^2+^, Na^+^, and Zn^2+^) were determined by inductively coupled plasma spectrometry (ICP/OES, Perkin-Elmer Optima™ 8300, Waltham, MA, USA) all quality control and assurance measures were taken, including calibration check measures, determination of the method’s limit of detection (MLD). The limit of detection (LOD) for each metal were determined as follows: 8 independent analyses of a blank solution spiked with the metal at a level of lower concentration of the analytical curve were performed. The LOD were calculated from the standard deviation (σ) of these determinations. A calibration curve was constructed using a standard multi-ionic solution at 0–30 ppm concentrations for Ca^2+^, K^+^, Mg^2+^, and Na^+^, and 0–10 ppm for B^+^, Fe^2+^, Mn ^2+^, and Zn^2+^. For intermediate standard solution were prepared using 2% HNO_3._ The linearity of the calibration curve based on the correlation coefficient, and all correlation coefficients were greater than 0.995. The ICP/OES method showed that a low levels of limit of detection (LODs) which were varied between (0.01–1 ppm).

### 2.8. In Vitro Protein Digestibility

The in vitro protein digestibility of unprocessed and processed flour seed was evaluated according to the official method (982.30.C) of the Association of Official Analytical Chemists [10]. Brefly, 0.26–0.36 g of sample containing 10 mg N, were soaked in 10 mL of water for 1h, the pH was adjusted to 8 and incubated at 37 °C. The enzyme solution, containing 227,040 BAEE units of porcine pancreatic trypsin, 1860 BAEE units of bovine pancreatic α-chymotrypsin, and 0.52 L-leucine β-nafphthylamide units of porcine intestinal peptinase (Sigma Chemical Co., St Louis, MO, USA) was added, adjusted to pH 8 and incubated for 10 min at 37 °C; then bacterial protease (Sigma Chemical Co., St Louis, MO, USA) was added, incubated at 20 °C for 20 min, and the final pH was measured. for comparison purposes, casein (Sigma Chemical Co., St Louis, MO, USA) was used as reference.

### 2.9. Anti-Nutritional Compounds

The anti-nutritional compounds most frequently found in amaranth seeds (lectins, tannins, trypsin inhibitors, saponins, and cyanogenic glycosides) were quantified in unprocessed, popped, and germinated seeds.

Lectin were extracted from the amaranth flour overnight with phosphate buffered saline (PBS) solution (10 mM, pH 7.4) at 4 °C with constant stirring in the ratio 1:10 (*w*/*v*) flour: buffer. Proteins were recovered by centrifugation at 10,000 rpm for 30 min. The lectins activity was determined in the supernatant, by two-fold serial dilution [12]. using human erythrocytes type O and A in a U-shaped microtiter 96-well. The lectin-containing solution was serially diluted (2-fold), the sample volume in each well was adjusted to 50 μL with PBS, and the diluted samples were mixed with 50 μL of the 2% suspension of human erythrocytes. The reaction mixture was incubated 1 h at room temperature, and then observed for agglutination. The titer was defined as the reciprocal of the highest dilution showing detectable agglutination.

Tannins were determined according to the modified vanillin-HCl method described by Price et al. [13] using (+)-catechin (Sigma Chemical Co., St Louis, MO, USA) as standard. Tannins were extracted from 0.4 g of flours with 10 mL methanol under agitation for 1 h at room temperature and centrifuged 30 min at 3500 rpm. The supernatant was reacted with vanillin-methanol:HCl solution (0.5:8 %) in acidified methanol and the absorbance was measured at 500 nm (Perkin Elmer Lambda 40 UV/Vis spectrometer). Tannin content was expressed as mg of catechin (CE) per gram of sample (mg CE/g). 

Trypsin inhibitory activity was measured as the residual trypsin activity using BAPNA (N-benzoyl-DL-arginine p-nitroanilide) [14]. Trypsin inhibitors were extracted from 1 g of sample with NaOH (0.01 N) at pH 9.6 for 2.5 h under agitation and the solids were separated by centrifugation for 30 min at 5000 rpm. To the supernatant containing the trypsin inhibitors, trypsin solution (20 µg /mL HCl 0.001N) was added, incubated 10 min at 37°C, BAPNA (0.4 mg/mL pH 8.2) (Sigma Chemical Co., St Louis, MO, USA) was added and reacted 10 min at 37 °C. The reaction was stopped with acetic acid (30%), and the absorbance was measured at 410 nm in a UV spectrophotometer (Perkin Elmer Lambda 40 UV/Vis spectrometer). Total trypsin inhibitory activity was expressed as trypsin inhibitory units (TIU) per mg of the sample (TIU/mg) on a dry weight basis. 

Saponins were extracted for 2 h from 7 g of the ground sample using 85:15 (%) methanol–water solution. The solvents were eliminated by rotary evaporation, and saponins extracted were diluted in NaCl solution (0.9%). The assay was carried out as described in pertinent literature [15] employing the serial dilution method, using human erythrocytes type O in a U-shaped microtiter 96-well. The solution containing saponins was serially diluted (2-fold), the sample volume in each well was adjusted to 50 μL with NaCl (0.9%), and the diluted samples were mixed with 50 μL of the 4% suspension of erythrocytes. The reaction mixture was incubated 1 h at room temperature and then observed maximum dilution showing hemolysis. The analyses were performed in triplicate and the results were reported as hemolytic units.

Cyanogenic glycosides were detected using the technique of the picrate-impregnated paper according to Butler [16] Briefly, a solution of β-glucosidase (5 mL, 1 mg/mL) (Sigma Chemical Co., St Louis, MO, USA) was added to 0.5 g of sample, ground at the time of the assay, the indicator paper is placed, without touching the reaction mixture, hermetically sealed and incubated 2 h at 40 °C. The appearance of a brown pigment on the paper is indicative of the presence of cyanides in the sample.

### 2.10. Statistical Analysis

An ANOVA was performed to determine the effects of traditional popping and germination treatments on the anti-nutritional compounds in amaranth seeds. In addition, ANOVA and Tukey’s test (*p* < 0.05) were conducted to determine the differences between the results obtained by proximal analysis, morphological analysis, and mineral determination. All statistical analyses were performed using StatGraphics Centurion version 19.1.2 (StatGraphics, The Plains, VA, USA).

## 3. Results

### 3.1. Seed Size and Weight

The summary statistics on the length, width, and weight of 100-seed samples obtained from the five regions in focus of this investigation are presented in Table 1. Significant variations were found among the samples, whereby MX1 and PU seeds were the widest and PR was 8.26% longer than MX2, whereas the seeds in the remaining three samples were of comparable length. Although EM and PU seeds were thicker than those from the other three samples, the difference was not statistically significant. Moreover, MX1 and PU had the highest weight, whereas MX2 had the lowest.

### 3.2. Proximate Analysis

Table 2 shows the findings of proximate analysis involving untreated seeds. The protein percentages ranged from 11.35% to 18.80%, with the highest values obtained for MX1 and MX2, whereas those obtained for PR, EM, and PU were comparable (*p* > 0.5) and were lower by 33.76%. The ethereal extract values ranged from 0.277% (EM) to 13.39% (MX2). On the other hand, no significant differences in crude fiber or moisture content were noted among samples. The ash content of PR and MX1 (3.32%) was comparable (*p* > 0.5), exceeding that measured for MX2, EM, and PU by 15.65%. In the case of carbohydrates, significant differences (29.24%, *p* < 0.5) were observed, with EM registering the highest and MX2 the lowest value.

### 3.3. GC-MS Determination of the Fatty Acid Profile

Table 3 shows the results of fatty acid concentrations in the untreated seeds. Linoleic and oleic acids stand out. Arachidonic acid was present in lower concentrations. In some samples, not all fatty acids were detected. The samples EM and PU showed all fatty acids of interest. Linoleic, linolenic, and arachidonic acids corresponding to the omega 3 and 6 fatty acids group, considered essential for health, were found in all samples.

### 3.4. Mineral Content Determination

The findings yielded by mineral content analyses are presented in Table 4, indicating significant differences (*p* < 0.5) among the five samples. Although Ca^2+^, K^+^, and Mg^2+^ predominated in all samples, K^+^ was present in the highest concentrations, ranging from 49.13 to 32.66 ppm. Moreover, B^3+^ content was the highest in PU, and the measured value exceeded that obtained for PR and MX1 by 27.01%. However, MX1 registered the highest Ca^2+^ and K^+^ values, whereas EM had the lowest value, with a difference of 38.64% and 33.52%, respectively. Fe^2+^ content in MX1 was 49.72% (*p* < 0.5) greater than in the other four samples which had similar amounts. On the other hand, Zn^2+^ (0.03 to 1 ppm) and Mn^2+^ (0.01 to 0.36 ppm) recorded the lowest concentrations. 

### 3.5. Digestibility

Digestibility is an essential parameter for assessing protein quality, which was determined in the amaranth seed meal, and the findings of in vitro assessments are shown in Figure 1. As expected, digestibility of untreated seeds was lower compared to popped and germinated seeds. The highest digestibility was noted in untreated EM seeds (*p* < 0.5), but the difference between the lowest and the highest values was only 3.3%.

After popping, PU and MX2 seeds exhibited the highest digestibility, whereas MX1 had the lowest value (with a 9.89% difference relative to PU). When germinated seeds were assessed, the lowest digestibility value was obtained for MX1 and the highest for PU (with 9.94% difference).

### 3.6. Anti-Nutritional Compounds

Anti-nutritional compounds were determined in untreated, germinated, and popped seeds to evaluate the effect of these treatments. One of the anti-nutritional compounds tested was lectins. Table 4 shows the hemagglutinating activity results of these proteins in the three treatments. 

Hemagglutination was observed in both types of erythrocytes used. When performing the test with type A erythrocytes using the untreated samples, the highest activity was obtained for PU and the lowest for MX1 (with an 82.76% difference between the two). MX1 showed the highest activity in the popped seeds, which was 50% higher than that measured in all other samples that had comparable results (*p* > 0.5). When the test with type O erythrocytes was carried out on untreated seeds, MX1 expressed the least activity, and the PU sample exhibited the highest activity (with a difference of 82.78%).

Lectins were still present after popping, whereby 1.37–14.81% of their biological activity remained in the samples. However, this type of protein considerably decreased after germination, and its biological activity ranged from 4.99% to 28.59% (recorded in the PU sample).

Tannins were detected in untreated as well as treated seeds. As shown in Table 4, the concentration in the untreated seeds was the highest in PU and was the lowest in MX1. After germination, the content of these compounds increased in all samples.

The results of trypsin inhibitor activity in treated and untreated seeds are shown in Table 5. These compounds were only found in raw seeds, and their concentration was the highest in MX2. Similarly, only untreated seeds contained saponins and, although the findings for MX1, MX2, EM, and PU were comparable, they were 50% higher than those obtained for the PR sample. 

## 4. Discussion

Although amaranth can be consumed in various forms, germinated and popped seeds are most commonly used in Mexico. In this study, the traditional popping method was adopted, even though the use of clay container makes it difficult to achieve uniform temperature and results in many damaged or unpopped seeds. In this study, the traditional popping method was adopted, even though the use of clay container makes it difficult to achieve uniform temperature and results in many damaged or unpopped seeds. The inevitable browning also reduces the nutritional quality of the seed, and decreases or eliminates essential amino acids and vitamins, as well as some other beneficial seed constituents, adversely affecting the quality of the final product [17]. The effects of these methods on the anti-nutritional components and the harvest yield were evaluated in this study.

Amaranth seed size varies among different species and growing conditions. Although all five samples included in the current analyses belonged to the same genus, they exhibited variability due to climate, soil, and nutrients prevalent in the growing area, impacting the grain size. The width of all samples was about 32.4–33.3% lower than the measurements reported by several authors. The five samples were also approximately 14.6–26.3% shorter and 9% thinner than those studied by other authors [18]. The weights obtained in this study (based on 100 seeds per sample) were similar to those reported by De Bock et al. in 2021 [19]. Other authors have reported highly variable values, ranging from 0.03 to 0.79 g, including species such as *Amaranthus hypochondriacus* [20,21,22]. However, in the present study, 6–32% higher seed weight was noted for MX1 and PU compared to the other three samples, which is consistent with their larger seed sizes. 

Proteins are essential nutrients, and are of great importance in human and animal diet. However, they must be adequately digested, and the same applies to amaranth proteins [2]. The protein values in different species of amaranth range between 13.56% and 17.25% [19,23,24]. In *Amaranthus hypochondriacus*, 16% protein content has been reported [23,24] while our analyses revealed 11.35% to 18.8%, differences may be partially or completely due to species, growing conditions, soil quality, and climate [25,26].

Lipids are essential nutrients for human health and were reported in *Amaranthus hypochondriacus* in the 5.6–7.8% range [19,21,24]. In this study, four samples showed lower levels than those reported, and MX2 had a higher content. Environmental factor is known to effect seed size of amaranth, and the seed size is associated with chemical composition, therefore it is possible that day length, growing temperature, water availability and other environmental factors affect fat content [1,27].

The fiber content in the five samples analyzed was higher levels previously reported by several authors (1.76–6.5%). However, only MX2 exceeded the reported results in terms of total carbohydrates. In turn, ash and humidity had similar percentages to those noted by other authors for this type of seed [23,24,28,29,30].

Amaranth seeds contain several minerals that are essential for human health, with higher P^+^, Ca^2+^, Fe^2+^, K^+^, and Mg^2+^ contents than in other grains such as maize, rice, and sorghum. Minerals are important constituents of human diet, as they serve as cofactors for many physiological and metabolic processes. Calcium is required for bone growth as well as in muscular and neurological functions, while iron is important for hemoglobin development [31].

The availability of calcium can be affected by some antinutritional compounds, trapping it and preventing its function, such as oxalates. In the Amaranthaceae family, it has been shown that oxalates are found in seeds, leaves, and stems, between 59–1029 mg/100 g, depending on the part of the plant. It has been shown that the consumption of *Amaranthus gangeticus* leaves significantly reduces the absorption and utilization of calcium, which was related to the oxalate content in the leaves [21].

The minerals with the highest concentration in the five samples studied were K^+^, Mg^2^, and Ca^2+^, which is in agreement with the findings reported in extant literature [32]. However, significant differences can occur due to genetic and geographic factors [33]. 

Fatty acids of nutritional interest are found in amaranth seeds. Fatty acids, carotenoids, and tocopherols are the three most important lipophilic nutrients found in plant-based foods. Fatty acids, especially the long-chain polyunsaturated fatty acids (PUFAs), are essential nutrients for human growth and development. PUFA contents influence both membrane structure and proper functions of membrane-associated proteins in a wide variety of tissues, including the brain and retina. In this study, unsaturated fatty acids were found in the analyzed samples [30]. Amaranth oil contains approximately 77% polyunsaturated fatty acids which are mostly found within the germ. The most representative ones are oleic, linoleic, linolenic, and arachidonic acids. These fatty acids belong to omega 3, 6, and 9 acids and are essential for human health. The results obtained for the five samples concur with those reported by other authors for oleic, linoleic, and linolenic acid content in other amaranth species (22.40–49.00, 35.31–47.00, 0.56–3.82%, respectively).

Linoleic acid was found in the highest amount in all samples. This acid exhibits physiological activities and acts as antiatherosclerotic, antihypercholesterolemic, anticarcinogenic, immunodulating, antidiabetic agent, metal-chelating, and lean body mass enhancer [34,35,36]. 

As the protein concentration in the analyzed amaranth samples was high, it was interesting to evaluate their digestibility [37] *In vitro* protein digestibility is a valuable parameter in the assessment of the nutritional potential of proteins [25]. Untreated amaranth seeds had 75–77.5% digestibility which was higher than the range reported for this type of seeds (35.8–74.2%) [38,39], but it is similar to raw seed flours of other amaranth species, ranging from 72–78% [40,41]. 

For species such as *Amaranthus cruentus* and *Amaranthus caudatus*, digestibility has been found to range between 80% and 82% and these percentages could be due to anti-nutritional compounds that, even in small concentrations, can interfere with protein digestibility [42]. The protein digestibility in the samples examined in the present study improved after popping by 0.5–8.7%. This is in line with the available evidence that popping affects digestibility, both In vitro and In vivo. However, heat treatment can decrease amaranth protein digestibility, as a result of several factors, including degradation of amino acids, affecting the levels of essential amino acids, promoting denaturation (which can affect the solubility of proteins, making digestibility difficult), as well as promoting the formation of protein interactions with other compounds, which does not allow enzymes to act appropriately, adversely affecting digestion [43,44]. 

Findings yielded by several studies indicate that digestibility can increase by 2% to 9.9% after trapping. As a result of this treatment, protein digestibility can reach up to 84%, and would thus represent a good supply of lysine [45,46]. In the present study, following germination, protein digestibility in amaranth seeds increased by 0.88–9%, which is comparable to 7% reported for *A. caudatus*, but is significantly below 55% measured for *A. viridis* [47].

Improvements in digestibility can be attributed to the fact that the amount of some anti-nutritional compounds decreases during germination, allowing proteins to be properly digested. However, a 0.5–14.4% decrease in digestibility after germination has been reported for amaranth seeds of various species [48]. Nonetheless, in vivo assays demonstrate an increase in protein quality after germination [2,48,49].

Presence of anti-nutritional compounds in treated and untreated seeds of several amaranth species has been reported in extant literature, as processing affects the content of some of these compounds [8,9]. In the present study, anti-nutritional compounds were found in all five amaranth seed samples. Lectins, for example, are found in a higher proportion in seeds These proteins can bind to cells, causing structural and physiological damage to tissues when consumed, with adverse health effects [50]. 

Lectin is present in amaranth seeds of various species in concentrations ranging from 0.5 to 7.3 g/100 g [51,52], supporting our findings. 

We also noted a 50–70% lower lectin activity in type A compared to type O erythrocytes. Pure amaranth lectins show a higher specificity towards N-acetyl-D-galactosamine (GalNAc) and the disaccharide Galβ1-3GalNAc [52,53].

GalNAc is found in the membrane of type A erythrocytes, which allows lectins to bind to erythrocytes. These lectins can bind, albeit with lower affinity, to monosaccharides such as galactose, galactosamine, and oligosaccharides containing GalNAc. Because these carbohydrates are found in the membrane of type A and O erythrocytes, it was possible to observe haemagglutination in both types of erythrocytes.

*Amaranthus leucocarpus syn hypochondriacus*, *Amaranthus caudatus*, and *Amaranthus spinosus* have been shown to exhibit comparable haemagglutinating activity with respect to all ABO system erythrocytes. Nonetheless, differences in haemagglutinating activity may occur due to several factors, including species characteristics.

In this study, lectin activity in the sample decreased by 71.1–95.0% and 70.0–98.6% after germination and popping, respectively, making the seeds much more nutritious. 

Heat treatment and germination affect the physicochemical properties of proteins. During germination, a loss of globulins and a decrease in the albumin/globulin fraction is observed after seed hatching and partial hydrolysis of the proteins, which explains the loss of haemagglutinating activity of the lectins. The decrease in lectin content of different seeds varies due to differences in seed genotypes and experimental conditions used [54]. Several studies have been published about cytotoxic damage to animal organs caused by lectins or even death [55,56]. Although some remnants were still found in this study after popping and germination, both treatments were highly effective in significantly reducing the biological activity of lectins. 

The consumption of lectins from other sources, such as beans and some cereals, can cause damage to the gastrointestinal system and can even cause fatal effects due to nutrient absorption. The level of toxicity among lectins varies widely, ranging from merely anti-nutritional properties to severe toxicity, as in the case of Ricinus communis and Phaseolus spp. lectins (LD50 0.05 and 1100 mg/kg body weight in mice respectively).

From the nutritional point of view, the toxicity of lectins depends on the quantity that is consumed and the frequency of consumption [55,57]. 

Trypsin inhibitors are compounds of protein origin found in a wide variety of cereal and legume seeds. The results obtained in the present work showed that they were present in low concentrations in untreated seeds. Some authors have reported the presence of these compounds in *Amaranthus hypochondriacus* seeds, in the 0.005–3.500 TUI/mg concentration range [58,59]. The concentrations obtained in the present study are higher, indicating influence of the location and plant growing conditions on the seed composition.

When comparing the trypsin inhibitor content of *Amaranthus hypochondriacus* with that reported for other amaranth species, important differences can be noted. In some cases, such as *Amaranthus muricatus*, these compounds are not detected [60]. In contrast, in *Amaranthus pumilus*, *Amaranthus anclancalius*, *Amaranthus caudatus*, and *Amaranthus cruentus*, the concentrations ranged between 0.0 and 10.8 TUI/mg [59,61]. Once again, the growing conditions and location are likely responsible for the differences between species.

In the present study, no trypsin inhibitor activity was detected in any of the samples after germination or popping. However, other authors have observed a significant decrease in the activity of these compounds in both germinated and popped *Amaranthus cruentus* and *Amaranthus caudatus* seeds [54]. This decrease during germination may be attributed to an increase in proteolytic enzyme activity, which promotes the degradation of these compounds that are used as an energy source for plant development [61,62]. On the other hand, the loss of activity after popping can be explained by the severe effect that temperature has on the structure of these protease inhibitors [63].

Saponins are triterpene-type anti-nutritional compounds that can be glycosylated and have the capacity to produce hemolysis of erythrocytes. Therefore, it is crucial to reduce or eliminate them from human diet to avoid any related health risks. These compounds were found in the five samples analyzed in this work, but were only detected in the untreated seeds at low concentrations (5.33–10.66 HU/mg). These findings are in agreement with those reported for *Amaranthus viridis*, *Amaranthus cruentus*, *Amaranthus anclancalius*, and *Amaranthus caudatus* seeds [47]. Although no hemolytic activity was detected in germinated and popped seeds in our analyses, in 1999, Oleszek and colleagues reported that after 240 h of germination, such compounds were still detected and even increased in concentration [64].

Tannins are polyphenolic compounds of plant origin present in all vascular plants, including legumes, cereals, and other plants used for food. These compounds have beneficial and detrimental health effects depending on their nature, concentration, animal species, and individual health status. In the present study, low tannin concentrations were detected only in the untreated seeds because this type of compound is related to the color. Since *Amaranthus hypochondriacus* seeds have a light golden color, the tannin content is low, as reported by De Bock and colleagues [19]. These authors measured low concentrations of phenolic compounds in various amaranth seeds, reporting 0.316–0.511 mg TAg/g for *Amaranthus caudatus* seeds. For *Amaranthus hypochondriacus* (white and golden type) and *Amaranthus hybrid* (white), these compounds were found in 5.4, 10.9, and 0.6–9.7 g/kg quantities, respectively [65].

The increase in phenolic compound content following heat treatment has been attributed to thermal degradation, bran detachment, the release of phenolic compounds bound to different cellular compartments, and formation of non-endogenous phenolic compounds in the seed. Extant evidence also suggests that it may also be due to the formation of thermal degradation products, such as Maillard reactions, phenol oxidation and maderization [66].

Due to the adverse effects, they can cause to health, tannins have been considered undesirable for human and animal consumption, since they can form complexes with proteins and digestive enzymes, decreasing the nutritional value of food because the proteins bound to tannins cannot be digested. In the case of enzymes, they lose their activity, affecting the digestion of food. Tannins can also cause darkening reactions in food, which would compromise the sensorial attributes.

The implication of tannins present in foodstuffs for human health may be of concern since they act beneficially (as pro- or anti-oxidants, cancer, and mutation preventive agent) as well as adversely (as cancer promoters or cause harmful nutritional effects). However, the amount of tannins consumed plays an essential role in these sequelae.

Cyanogenic glycosides are compounds that can represent a high health risk if consumed, but were not detected in any samples studied before or after the treatments. These results concur with those reported for several amaranth species [67]. The absence of cyanogenic glycosides in the samples may be attributed to these compounds being preferentially found in certain families, such as Fabaceae, Rosaceae, Leguminosae, Linaceae, and Compositae [65].

## 5. Conclusions

Amaranth seeds represent a valuable source of nutrients, such as minerals, omega-3 fatty acids, and proteins. The treatments to which the amaranth seeds from the five localities were subjected improved the in vitro digestibility of proteins, for which germination was particularly beneficial.

Both popping and germination decreased the amounts of anti-nutritional compounds of protein origin and saponins. Trypsin inhibitors and saponins were not detected after either treatment, whereas lectins decreased significantly, especially after germination. The increased tannin content may be due to physical and chemical factors rather than metabolic processes in the seeds.

These seeds have good digestibility, which is improved by germination and popping. Both treatments promote a decrease in anti-nutritional compounds, which would ensure better utilization of this type of food, reducing the health risk of anti-nutritional compounds. The results reported here show that germination and popping positively affect the quality of *Amaranthus hypochondriacus* by eliminating and reducing anti-nutritional compounds and improving digestibility, thereby enhancing the overall quality of this pseudo-cereal.

## Figures and Tables

**Figure 1 foods-11-02075-f001:**
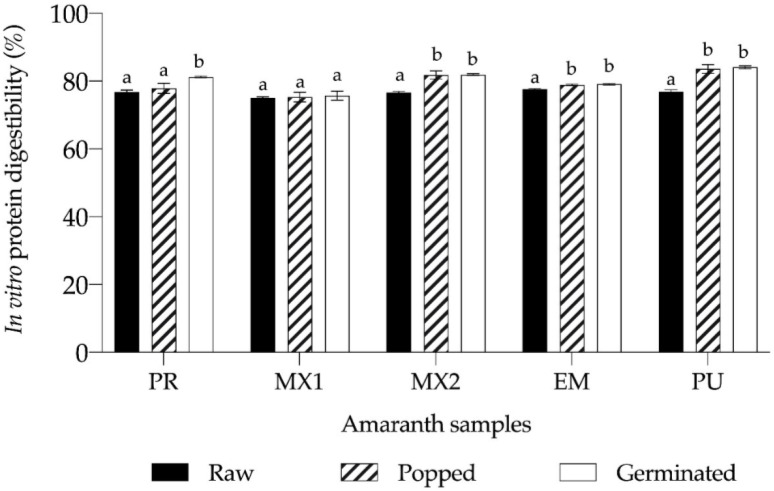
In vitro digestibility of raw, popped, and germinated *Amaranthus hypochondriacus* seeds. Data shown represent mean ± SEM obtained in at least three independent experiments. Means with different superscripts are significantly different (Tukey’s test, *p* < 0.05). Sample labels correspond to different planting locations, i.e., PR = Progreso, MX1 = Mixquiahuala 1, MX2 = Mixquiahuala 2, EM = Estado de Mexico, and PU = Puebla.

**Table 1 foods-11-02075-t001:** Weight and size of *Amaranthus hypochondriacus* seeds. Results are mean ± standard deviation (*n* = 3).

Samples	Dimensions (mm)			Weight (g/100 Seeds)
	Width	Length	Thickness	
PR	0.94 ± 0.069 ^b^	1.21 ± 0.128 ^b^	0.790 ± 0.034 ^a^	0.69 ± 0.157 ^a^
MX1	0.95 ± 0.07 ^b^	1.17 ± 0.048 ^ab^	0.792 ± 0.023 ^a^	1.03 ± 0.17 ^c^
MX2	0.87 ± 0.054 ^a^	1.11 ± 0.031 ^a^	0.791 ± 0.013 ^a^	0.62 ± 0.063 ^d^
EM	0.92 ± 0.064 ^a^	1.160 ± 0.107 ^ab^	0.821 ± 0.015 ^b^	0.69 ± 0.128 ^d^
PU	0.94 ± 0.064 ^b^	1.180 ± 0.079 ^ab^	0.822 ± 0.019 ^b^	0.95 ± 0.195 ^b^

Data shown represent mean ± SEM obtained in at least three independent experiments. Means with different superscripts are significantly different (Tukey’s test, *p* < 0.05). Sample labels correspond to different planting locations, i.e., PR = Progreso, MX1 = Mixquiahuala 1, MX2 = Mixquiahuala 2, EM = Estado de Mexico, and PU = Puebla.

**Table 2 foods-11-02075-t002:** Proximate analysis of raw *Amaranthus hypochondriacus* seeds.

Samples (%)	Protein	Ether Extract	Crude Fiber	Moisture	Ash	Carbohydrates
PR	12.29 ± 1.31 ^a^	2.98 ± 0.04 ^a^	7.17 ± 0.41 ^a^	10.55 ± 0.07 ^a^	3.34 ± 0.1 ^a^	63.65 ± 1.287 ^a^
MX1	18.80 ± 2.2 ^b^	4.67 ± 0.014 ^b^	7.35 ± 0.03 ^a^	10.37 ± 0.15 ^a^	3.29 ± 0.04 ^a^	55.50 ± 1.822 ^b^
MX2	18.63 ± 1.99 ^b^	13.39 ± 2.03 ^c^	7.01 ± 0.07 ^a^	10.53 ± 0.06 ^a^	2.96 ± 0.13 ^b^	47.45 ± 0.507 ^b^
EM	11.35 ± 1.05 ^a^	0.26 ± 0.01 ^d^	8.09 ± 1.58 ^a^	10.48 ± 0.1 ^a^	2.73 ± 0.15 ^b^	67.06 ± 0.595 ^d^
PU	13.55 ± 1.17 ^a^	0.27 ± 0.3 ^d^	10.85 ± 0.78 ^a^	10.50 ± 0.41 ^a^	2.70 ± 0.23 ^b^	62.10 ± 0.422 ^a^

Data shown represent mean ± SEM obtained in at least three independent experiments. Means with different superscripts are significantly different (Tukey’s test, *p* < 0.05). Sample labels correspond to different planting locations, i.e., PR = Progreso, MX1 = Mixquiahuala 1, MX2 = Mixquiahuala 2, EM = Estado de Mexico, and PU = Puebla.

**Table 3 foods-11-02075-t003:** Fatty acids content in raw *Amaranthus hypochondriacus* seeds.

Fatty Acid Concentrations %	Amaranth Samples				
	PR	MX1	MX2	EM	PU
Myristic (14:0)	ND	0.63 ± 0.06 ^b^	1.14 ± 0.14 ^a^	0.38±0.07^c^	0.19 ± 0.01 ^c^
Myristoleic (14:1)	ND	ND	ND	0.16±0.01^a^	0.16 ± 0.04 ^a^
Palmitic (16:0)	17.15 ± 0.95 ^a^	18.61 ± 1.44 ^a^	15.45 ± 1.45 ^a^	17.00±2.47^a^	16.01 ± 2.58 ^a^
Palmitoleic (16:2)	ND	ND	ND	0.8 ±0.06^a^	0.89 ± 0.02 ^a^
Palmitolenic (16:3)	ND	ND	ND	0.85±0.12^a^	0.92 ± 0.06 ^a^
Stearic (18:0)	11.9 ± 1.61 ^a^	3.47 ± 0.16 ^b^	1.81 ± 0.11 ^b^	2.80±0.40^b^	3.08 ± 1.02 ^b^
Oleic (18:1)	31.60 ± 0.35 ^a^	29.27 ± 1. 98 ^a^	31.51 ± 1.49 ^a^	29.41±0.66^a^	29.60 ± 1.29 ^a^
Linoleic (18:2)	45.96 ± 0.32 ^a^	45.33 ± 0.39 ^a^	46.05 ± 1.79 ^a^	45.78±1.09^a^	46.45 ± 2.30 ^a^
Linolenic (18:3)	1.56 ± 0.41 ^a^	1.14 ± 0.03 ^a^	1.43 ± 0.05 ^a^	1.22±0.20^a^	1.16 ± 0.22 ^a^
Arachidonic (20:4)	1.02 ± 0.12 ^a^	0.51 ± 0.19 ^b^	0.87 ± 0.07 ^c^	1.19±0.36^b^	0.83 ± 0.06 ^c^

Data shown represent mean ± SEM obtained in at least three independent experiments. Means with different superscripts are significantly different (Tukey’s test, *p* < 0.05). Sample labels correspond to different planting locations, i.e., PR = Progreso, MX1 = Mixquiahuala 1, MX2 = Mixquiahuala 2, EM = Estado de Mexico, and PU = Puebla. ND = Not detected.

**Table 4 foods-11-02075-t004:** Mineral content in raw *Amaranthus hypochondriacus* seeds.

Minerals (ppm)	Amaranth Samples					LOD (mg L^−1^ )	Slopes of the Calibration Curves
	PR	MX1	MX2	EM	PU		
B^3+^	5.30 ± 1.07 ^a^	5.39 ± 1.16 ^a^	5.99 ± 1.76 ^ab^	6.61 ± 2.38 ^bc^	7.33 ± 3.1 ^c^	0.970	0.9999
Ca^2+^	12.26 ± 0.012 ^b^	16.40 ± 0.006 ^c^	11.80 ± 0.014 ^b^	10.06 ± 0.016 ^a^	11.65 ± 0.004 ^b^	0.087	0.9998
Fe^2+^	0.34 ± 0.021 ^a^	0.70 ± 0.043 ^b^	0.33 ± 0.03 ^a^	0.36 ±0.088 ^a^	0.38 ± 0.014 ^a^	0.015	0.9998
K^+^	43.10 ± 4.2 ^b^	49.13 ± 3.88 ^c^	44.96 ± 2.4 ^bc^	32.66 ± 0.006 ^a^	32.98 ± 0.005 ^a^	0.025	0.9999
Mg^2+^	21.23 ± 1.38 ^ab^	25.03 ± 1.88 ^c^	23.23 ± 1.31 ^bc^	20.40 ± 0.043 ^a^	20.25 ± 0.065 ^a^	0.034	0.9999
Mn^2+^	0.01 ± 0.11 ^a^	0.06 ± 0.017 ^ab^	0.11 ± 0.08 ^b^	0.31 ± 0.04 ^c^	0.36 ± 0.066 ^c^	0.011	0.9998
Na^+^	2.28 ± 1.2 ^a^	1.75 ± 0.028 ^a^	1.49 ± 0.023 ^a^	1.49 ± 0.055 ^a^	1.46 ± 0.005 ^a^	0.045	0.9997
Zn^2+^	0.09 ± 0.011 ^a^	0.06 ± 0.017 ^a^	0.03 ± 0.08 ^a^	0.02 ± 0.04 ^a^	0.07 ± 0.006 ^a^	0.023	0.9997

Data shown represent mean ± SEM obtained in at least three independent experiments. Means with different superscripts are significantly different (Tukey’s test, *p* < 0.05). Sample labels correspond to different planting locations, i.e., PR = Progreso, MX1 = Mixquiahuala 1, MX2 = Mixquiahuala 2, EM = Estado de Mexico, and PU = Puebla.

**Table 5 foods-11-02075-t005:** Anti-nutritional compounds of raw, popped, and germinated *Amaranthus hypochondriacus* seeds.

Units	Treatment	*Amaranthus hypochondriacus* Resources				
		PR	MX1	MX2	EM	PU
Lectins concentration Erythrocytes Type A(HAU/mg protein)	Raw	91.03 ± 0.99 ^a^	11.24 ± 0.14 ^b^	31.50 ± 0.14 ^c^	59.19 ± 0.81 ^d^	65.19 ± 2.45 ^e^
	Popped	1.71 ± 0.012 ^b^	3.32 ± 0.013 ^d^	1.74 ± 0.0.32 ^bc^	1.66 ± 0.025 ^a^	1.78 ± 0.017 ^c^
	Germinated	6.51 ± 0.011 ^bc^	3.25 ± 0.005 ^a^	6.52 ± 0.013 ^c^	6.49 ± 0.016 ^b^	6.52 ± 0.015 ^bc^
Lectins concentration Erythrocytes Type O(HAU/mg protein)	Raw	22.77 ± 0.25 ^a^	22.48 ± 0.29 ^a^	31.51 ± 0.14 ^b^	118.38 ± 1.62 ^c^	130.58 ± 1.09 ^d^
	Popped	1.72 ± 0.012 ^b^	3.33 ± 0.014 ^d^	1.77 ± 0.032 ^bc^	1.63 ± 0.025 ^a^	1.79 ± 0.0177 ^c^
	Germinated	6.51 ± 0.01 ^bc^	3.26 ± 0.006 ^a^	6.25 ± 0.014 ^c^	6.49 ± 0.014 ^b^	6.52 ± 0.015 ^bc^
Tannins concentration (mg catechin/g)	Raw	0.105 ± 0.005 ^ab^	0.086 ± 0.004 ^a^	0.0933 ± 0.038 ^ab^	0.210 ± 0.033 ^c^	0.111 ± 0.001 ^b^
	Popped	1.671 ± 0.051 ^bc^	1.517 ± 0.034 ^a^	1.5967 ± 0.070 ^ab^	1.802 ± 0.051 ^c^	1.969 ± 0.032 ^d^
	Germinated	1.554 ± 0.036 ^b^	1.907 ± 0.065 ^c^	1.9592 ± 0.008 ^c^	1.407 ± 0.006 ^a^	1.382 ± 0.083 ^a^
Trypsin inhibitors (UTI/mg)	Raw	0.831 ± 0.123 ^ab^	0.868 ± 0.01 ^ab^	1.016 ± 0.43 ^b^	0.848 ± 0.05 ^ab^	0.519 ± 0.08 ^a^
	Popped	ND	ND	ND	ND	ND
	Germinated	ND	ND	ND	ND	ND
Saponins (HU/mg)	Raw	5.33 ± 0 ^a^	10.66 ± 0 ^b^	10.66 ± 0 ^b^	10.66 ± 0 ^b^	10.66 ± 0 ^b^
	Popped	ND	ND	ND	ND	ND
	Germinated	ND	ND	ND	ND	ND

Results are mean ± standard deviation (*n* = 3). Different letters in the same raw indicate significantly different reported values (*p* < 0.05). PR = Progreso, MX1 = Mixquiahuala 1, MX2 = Mixquiahuala 2, EM = Estado de Mexico, PU = Puebla. ND = Not detected.

## Data Availability

Data is contained within the article.

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
