# Peer review of "Effects of Germination and Popping on the Anti-Nutritional Compounds and the Digestibility of Amaranthus hypochondriacus Seeds"

_foods, 2022, doi:10.3390/foods11142075_

Round 1

Reviewer 1 Report

Changes and corrections to be made by the authors are highlighted in the manuscript

Author Response

Dear Reviewer 1.

We attach the file with the pertinent corrections.

Greetings

  1. Temperature is an important parameter. Therefore, it is necessary to indicate, what is the temperature that is reached during popping of amaranth?

Answer 1: In the paper, the temperature at which the poppining was performed was added.

2.- How do you define the end point of germination? How many hours did the grain take to germinate?

Answer 2: Germination time and germination end point were added to the paper. “ seeds   were germinated by  72 h, rinsed daily with boiled water, and  were considered germinated once the radical was 2.5–3 mm long,”

3.- What is the particle size after grinding?  

Answer 3: Added information was added in the paper  "a grinder Cole Parmer (model 4301-00) was used to grind unprocessed and processed grains into flour, them sieved  (USA Standard testing Sieve)"

4.- Describe the method by which you evaluated protein digestibility.

Answer 4: we have already added the “In vitro digestibility “in the methodology section.

  1. Cite in order.

Answer 5: fatty acids were listed in ascending order of fatty acids

6.This corresponds to the fatty acid profile of the raw grain-to correct.

Answer 6: the title of the table has been corrected.

  1. Digestibillity of what?

Answer 7:  The title of the section was corrected for “In vitro digestibility”.

  1. Explain what accounts for the wide variety in protein content.

Answer 8: we had an error and it has been corrected, and an explanation has been given as to why the differences may occur.

  1. Explain the predominant factors influencing fat content.

Answer 9: The explanation was given as to why the differences in fat content may occur.

  1. Delete the word very. There are vegetables that have higher protein content that amaranth.

Answer10: The word very was changed

  1. What is the safe level of lectins for consumption?

Answer 11: information has been added

  1. Calcium content in grains is high. Is must be ensured that this nutrient is not found in the form of oxalate, through the analysis of oxalates.

Answer 12: The analysis of oxalates was not considered in the objectives of the paper; however, because there are papers that report on the content of these compounds in amaranth, and explain their effect on bioavailability and their possible relationship in the formation of kidney stones, we have added information in the paper referring to these papers.

Reviewer 2 Report

In general, the manuscript is very interesting and may have some practical meaning. However, especially methodological part was very poorly described. It is definitely not a scientific level. Below, are my recommendations for improving the manuscript:

1. Fragment 64-73 - compounds indicated in the manuscript like phenolics, tannins, phytic acid, oxalates etc. are rather non-nutritive substances. To be precise, anti-nutritive (or anti-nutritional, both forms are accepted) are substances that hinder the use of other nutrients. On the other hand, all substances that are not used by the body, but are present in food, are non-nutritive substances, which also include anti-nutritional substances.  Therefore I propose to change the title of the article "anti-nutritional" to "non-nutritional". The group of non-nutritional substances is very wide, and also contains substances, which may be harmful to the body, like toxic heavy metals or nitrates. For details you can reffer to a recent paper: https://doi.org/10.3390/nu14081626

2. I suggest to add a separate chapter "Chemicals" and describe there all reagents in details.

lines 129-130 - what does it mean ICP/O? was it ICP-OES or MS? I think the first, but it must be clearly indicated.

3. Minerals determination - whether this method was previously standardized. Please provide appropriate citation or provide more details of the methodology used.

line 109 - please provide exact number of AOAC method used for Proximate Analysis

4. Table 2 - if the ether extract is identical to the fat content, please indicate that in the table.

5. Please describe sampling in details. How many seeds were taken to each analysis, and what were the samples (weights for each analysis).

6. lines 221-226 - how the digestibility was determined. You cannot write in the scientific paper that " it was determined in the amaranth seed meal" just like that. You must describe how was it performed.

Author Response

Dear reviewer 2.

We attach the file with the pertinent correction.
greetings
  1. Fragment 64-73 - compounds indicated in the manuscript like phenolics, tannins, phytic acid, oxalates etc. are rather non-nutritive substances. To be precise, anti-nutritive (or anti-nutritional, both forms are accepted) are substances that hinder the use of other nutrients. On the other hand, all substances that are not used by the body, but are present in food, are non-nutritive substances, which also include anti-nutritional substances.  Therefore I propose to change the title of the article "anti-nutritional" to "non-nutritional". The group of non-nutritional substances is very wide, and also contains substances, which may be harmful to the body, like toxic heavy metals or nitrates. For details you can reffer to a recent paper: https://doi.org/10.3390/nu14081626
Answer 1: only those compounds that are considered to interfere with the absorption of nutrients or that may cause adverse health effects are being considered, therefore the name "anti-nutritional" has been maintained.

 2.I suggest to add a separate chapter "Chemicals" and describe there all reagents in details.

Answer 2: The chapter about reagents used in the paper has been added.

3. lines 129-130 - what does it mean ICP/O? was it ICP-OES or MS? I think the first, but it must be clearly indicated.

Answer 3: the equipment has been corrected and relevant information has been added.

4. Minerals determination - whether this method was previously standardized. Please provide appropriate citation or provide more details of the methodology used.

 Answer 4: added information on the methodology used for the determination of metals by ICP.

5. line 109 - please provide exact number of AOAC method used for Proximate Analysis

Answer 5. Each determination in which AOAC methods were used has the number of the methodology in parentheses.

6.Table 2 - if the ether extract is identical to the fat content, please indicate that in the table.

Answer 6: The table was corrected; the name of the column was changed to Fat.

7. Please describe sampling in details. How many seeds were taken to each analysis, and what were the samples (weights for each analysis).

Answer 7.  The description of the methods used was expanded. The weight of samples was indicated, and information on the methodologies was added.

8. lines 221-226 - how the digestibility was determined. You cannot write in the scientific paper that " it was determined in the amaranth seed meal" just like that. You must describe how was it performed.

Answer 8. We apologize for the error of omitting of the digestibility methodology, we have already added the “In vitro digestibility “in the methodology section.

Round 2

Reviewer 1 Report

In the abstract the antinutrients of amaranth should be emphasized, since this is the objective of the study.

Author Response

Dear reviewer:

Information on anti-nutritional compounds has been added, in addition to adding more references.  

Reviewer 2 Report

The apaper was significantly improved, but it still needs some corrections.

1. lines 64-67 - once again I ask to clearly indicate that not all compounds mentioned in this fragment are antinutritional, because it is not true. Therefore I ask to change this fragment as follows: "Available evidence further indicates that amaranth seeds may contain some anti- and non-
nutritional compounds".

2. Once again I ask to provide appropriate citation regarding mineral's determination. It was not corrected. We are not sure if this method was previously validated and checked for its accuracy and precission. lines 143-144 - if these parameters were checked, so please provide appropriate results for LOD, LOQ, precission etc.

Author Response

Dear Editor, in response to the requested observations

1. Lines 64-67 - again I request that it be clearly stated that not all compounds mentioned in this fragment are antinutritional, because it is not true. I therefore request to change this fragment as follows: "The available evidence further indicates that amaranth seeds may contain some anti and non
nutritional".

The suggested text is added, in addition to adding two more bibliographic references.  Marked in another color 

2. Once again I request that the mineral determination be properly cited. It has not been corrected. We are not sure if this method was previously validated and checked for accuracy and precision. lines 143-144 - if these parameters were checked, then please provide appropriate results for LOD, LOQ, precision etc.

Information added. For mineral detection, detection limit was performed for each mineral values are added in table 4. Calibration curve was performed, also for each metal, results added in table 4.  
